# Predicting the potential suitable habitats of *Rosa roxburghii* and its key pest *Grapholita molesta* in China using the MaxEnt Model

Hongling Qin ᴰ*, Shuangshuang Wu, Guoqing Luo, Meifang Lu, Heng Xiang, Zhengbin Li

Guiyang Institute of Humanities and Technology, Guiyang, China

* qinhongling726@163.com

## Abstract

Climate change can reshape the potential distributions of crops and their pests, as well as their spatial overlap, with important implications for agricultural security and pest management. *Rosa. roxburghii*, a key specialty crop in southwestern China, is threatened by its primary pest, *Grapholita molesta*. Here, we used occurrence records and WorldClim bioclimatic variables to model the potential distributions of both species under current (1970–2000) and future (SSP2–4.5 and SSP5–8.5, 2061–2080) climates via a parameter-optimized MaxEnt model, and analyzed spatial overlap across suitability levels. Model performance was assessed with ENMeval and the continuous Boyce index. Results show that the minimum temperature of the coldest month (bio6) is the main climatic constraint for both species, with *R. roxburghii* influenced by multiple factors and *G. molesta* primarily limited by winter low-temperature thresholds. Currently, *R. roxburghii*'s suitable areas are smaller and mainly concentrated in southwestern and central China, whereas *G. molesta* has a much broader potential distribution across China. Although the overall co-occurrence area spans most of the suitable range of *R. roxburghii*, the majority of this overlap occurs in low-suitability zones, indicating a generally low probability of co-occurrence across much of the distribution. The overlap of highly suitable areas (HSA) between the two species is extremely limited, covering only ~28,800 km² (~0.3% of China's land area), suggesting that potential high-risk areas are spatially restricted rather than widespread. Under future scenarios, *R. roxburghii*'s highly suitable areas are projected to contract and fragment, whereas *G. molesta*'s range expands northward. However, the overlap of moderate and highly suitable areas between the two species does not increase and even declines, suggesting that future pest risk is likely to become spatially restructured, concentrating in specific regions rather than intensifying across the entire distribution range. These findings elucidate differences in climatic niche and climate sensitivity between crop and pest, providing a scientific basis for targeted pest management and optimized *R. roxburghii* cultivation.

**Data availability statement:** The climate data used in this study are available in the public database WorldClim (https://worldclim.org/). In addition, we also provide the future climate data and analysis code used in this study, which can be accessed at: https://doi.org/10.5281/zenodo.18870566.

**Funding:** This research was funded by the 2024 Scientific Research Foundation of Guiyang Institute of Humanities and Technology (Grant No. 2024rwscjs03). The funders had no role in study design, data collection and analysis, decision to publish, or preparation of the manuscript.

**Competing interests:** The authors have declared that no competing interests exist.

## Introduction

Herbivory by pests on economically important crops poses a major challenge to agricultural development, as it not only affects crop growth and yield but also constrains the geographical distribution of these crops [1,2]. Climate plays a pivotal role in determining species' ecological niches and thus critically influences the coexistence dynamics between pests and crops [3–5]. Under favorable climatic conditions, pests and crops are more likely to co-occur in the same regions, which often correspond to areas of intensified herbivory and represent key constraints on agricultural productivity [6].

*Rosa roxburghii*, a distinctive species in the Rosaceae family, has been widely cultivated in regions such as Guizhou due to its unique nutritional and economic value [7–10]. However, its growth and reproduction are frequently threatened by pests, resulting in substantial economic losses [11,12]. Among these pests, *Grapholita molesta* represents a primary threat. Originating in East Asia and now widespread across temperate and subtropical regions, *G. molesta* is a significant fruit tree pest reported in major production areas in China and worldwide [13,14]. This species has a broad host range, affecting peach, pear, apple, and other fruit crops, and exhibits strong ecological adaptability [15]. Its spread is facilitated not only by adult dispersal but also by human activities such as fruit trade and nursery plant transportation [16]. Previous studies have shown that winter low temperatures are a critical climatic factor limiting overwintering survival and regional establishment of *G. molesta*, with pest intensity varying significantly across different climatic regions [17,18]. *G. molesta* undergoes prolific reproduction during its active seasons, inflicting significant herbivory on *R. roxburghii* by consuming young leaves and fruits [15]. Larval boring into fruits results in rot and premature fruit drop, markedly decreasing fruit quality and yield [15,19].

Although research on *R. roxburghii* and *G. molesta* has progressed, most studies focus on pest prevention and control, while relatively few address the ecological niche distributions of both species or the spatial patterns of high herbivory risk [12,20]. Consequently, comprehensive understanding of the drivers and spatial extent of pest damage remains limited, hindering efforts in species conservation and pest management. Moreover, ongoing climate change is likely to significantly alter species' ecological niches and the distribution of herbivory risk [21–23]. Therefore, investigating the potential distributions and overlap of *R. roxburghii* and *G. molesta* under current and future climatic conditions is essential for informing long-term conservation and pest management strategies.

Building on this context, the present study focuses on *R. roxburghii* and *G. molesta* to: (1) map the potential suitable habitats of both species and assess their changes under current and projected future climates; and (2) examine the distribution and dynamics of their ecological niche overlap under these climatic scenarios.

## Materials and methods

### Species distribution data

In this study, distribution data for *R. roxburghii* and *G. molesta* were obtained from public databases and published literature. No field sampling was conducted, and

no research activities requiring specific site access or collection permits were involved. Therefore, no field permits were required. Distribution records of *R. roxburghii* were primarily collected from public databases, including the Global Biodiversity Information Facility (GBIF; https://www.gbif.org), the Chinese Virtual Herbarium (CVH; https://www.cvh.ac.cn), and the China Plant Photo Bank (http://www.plantphoto.cn). Compared with *R. roxburghii*, distribution records of *G. molesta* in China are relatively scarce in public databases. Therefore, a systematic literature search was conducted to supplement its distribution information. Literature searches were performed in the Web of Science (WOS; https://www.webofscience.com) and China National Knowledge Infrastructure (CNKI; https://www.cnki.net) databases using "*Grapholita molesta*\*" as the keyword. Considering that the climate data used in this study cover the period from 1970 to 2000, the literature search was restricted to publications from 1970 to the present. A total of 2,772 publications were retrieved, including 786 from WOS and 1,986 from CNKI. As a substantial proportion of these studies focused mainly on pest control, pesticide efficacy, or laboratory experiments and did not provide natural distribution information suitable for spatial analysis, further screening was required to improve the relevance and accuracy of distribution data extraction. Based on clearly defined inclusion and exclusion criteria, 36 publications containing explicit natural distribution records of *G. molesta* were ultimately retained for subsequent analyses (S1 Table).

**Inclusion criteria.** (1) The study explicitly focused on *G. molesta*; (2) The study content was related to species geographic distribution, population characteristics, ecological adaptability, or climatic influences. This criterion was applied to exclude irrelevant studies lacking natural distribution information from a large number of retrieved publications, thereby improving screening efficiency and ensuring accurate identification and extraction of studies containing explicit distribution records.

**Exclusion criteria.** (1) Studies focusing solely on pest control techniques, pesticide or efficacy trials, or laboratory-based physiological experiments without providing natural distribution information; (2) Studies involving *G. molesta* but lacking explicit descriptions of occurrence locations or distribution regions; (3) Review articles, conference abstracts, or duplicate publications that did not provide new or independently extractable distribution records.

Subsequently, the obtained species distribution data were subjected to the following basic filtering procedures: (1) retaining only records located within China; (2) removing records with zero latitude or longitude values; (3) excluding records with coordinates located at national capitals; (4) removing duplicate coordinate records; (5) excluding records with coordinates located at large museums or research institutions, where collection locations may have been erroneously recorded as institutional addresses; and (6) removing records located in marine areas. After these steps, 500 distribution records for *R. roxburghii* and 213 records for *G. molesta* were obtained.

To reduce the potential effects of spatial clustering and overfitting in species distribution models, Moran's I was used to quantify spatial autocorrelation in the distribution data for each species. Records with spatial autocorrelation test p-values < 0.1 were removed, and the remaining records with lower spatial autocorrelation were retained for subsequent analyses. Ultimately, a total of 526 distribution records were retained, including 373 records for *R. roxburghii* and 153 records for *G. molesta* (Fig 1, S1 Table).

The China base map used in this study was obtained from the Standard Map released by the National Platform for Common GeoSpatial Information Service of China (https://www.tianditu.gov.cn/, accessed in July 2025). This Standard Map was compiled in accordance with cartographic standards for national boundaries of China and other countries worldwide and is permitted for public uses such as news dissemination, book and journal illustrations, and the presentation of scientific research results. The map is freely accessible to the public for viewing and downloading.

## Climate data acquisition and processing

Current and future climate data were obtained from the WorldClim website (https://worldclim.org) using the R package geodata. Current climate data (1970–2000) consisted of 19 bioclimatic variables (Table 1), while future climate data were derived from the ACCESS-CM2 model of the CMIP6 project for 2061–2080 under two emission scenarios: SSP2–4.5

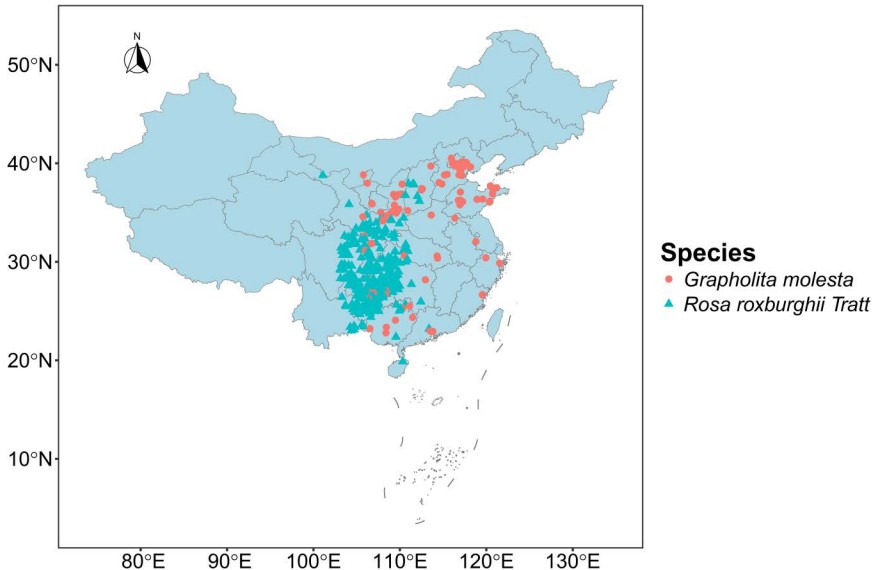

**Fig 1. Sample distribution of *R. roxburghii* and *G. molesta*.** Blue triangles indicate occurrence sites of *R. roxburghii*, and red circles indicate occurrence sites of *G. molesta*. Map Review Number: GS(2019)1822.

(medium emissions) and SSP5–8.5 (high emissions). All climate layers were obtained at a spatial resolution of 2.5 arc-minutes.

To reduce computational load and evaluate correlations among the bioclimatic variables, 10,000 random points were sampled across the study area, and the corresponding values for each climate variable were extracted for subsequent analysis. These randomly generated background points were used to characterize the overall climatic conditions of the study area, rather than relying solely on climate values at species occurrence locations, thereby avoiding potential sampling bias and ensuring that variable selection reflected general climatic correlations.

To avoid multicollinearity and ensure model stability, a two-step variable selection procedure was applied. First, Pearson correlation coefficients were calculated among all 19 bioclimatic variables, and variable pairs with $|r| > 0.7$ were considered highly correlated. Among these correlated variables, variables with clear ecological or physiological significance were prioritized; specifically, the minimum temperature of the coldest month (bio6) was retained, as it has been widely recognized as a key climatic factor influencing species survival and overwintering performance, while other highly correlated variables were removed [24–26]. Second, Variance Inflation Factor (VIF) analysis was performed on the remaining variables to further control for multicollinearity, and only variables with VIF < 5 were retained for model construction. Following this procedure, six climate variables were ultimately selected: bio2, bio6, bio14, bio15, bio18, and bio19.

### Ecological niche modeling, parameter optimization, and model evaluation

The potential suitable habitats of *R. roxburghii* and *G. molesta* were modeled using the Maximum Entropy (MaxEnt) approach, implemented via the maxnet package in R. Model parameter tuning and evaluation were conducted using the ENMeval package. Feature classes were tested in three combinations: linear (L), linear plus quadratic (LQ), and linear plus quadratic plus hinge (LQH), while the regularization multiplier (RM) was varied from 1 to 4 in increments of 0.5. To reduce the influence of spatial autocorrelation on model evaluation, a spatial block method was applied for cross-validation. The optimal model was selected based on the corrected Akaike Information Criterion for small sample sizes (AICc), with ΔAICc = 0 used as the selection criterion. Using the optimal model, continuous habitat suitability maps were

**Table 1. The 19 bioclimatic variables and their descriptions.**

| Bio Factors | Description | Unit | Choose or not |
|---|---|---|---|
| bio1 | Annual Mean Temperature | °C | |
| bio2 | Mean Diurnal Range (Mean of monthly (max temp – min temp)) | °C | * |
| bio3 | Isothermality (bio2/bio7) (×100) | - | |
| bio4 | Temperature Seasonality (standard deviation ×100) | °C | |
| bio5 | Max Temperature of Warmest Month | °C | |
| bio6 | Min Temperature of the Coldest Month | °C | * |
| bio7 | Temperature Annual Range (bio5-bio6) | °C | |
| bio8 | Mean Temperature of Wettest Quarter | °C | |
| bio9 | Mean Temperature of Driest Quarter | °C | |
| bio10 | Mean Temperature of Warmest Quarter | °C | |
| bio11 | Mean Temperature of Coldest Quarter | °C | |
| bio12 | Annual Precipitation | mm | |
| bio13 | Precipitation of Wettest Month | mm | |
| bio14 | Precipitation of Driest Month | mm | * |
| bio15 | Precipitation Seasonality (Coefficient of Variation) | - | * |
| bio16 | Precipitation of Wettest Quarter | mm | |
| bio17 | Precipitation of Driest Quarter | mm | |
| bio18 | Precipitation of Warmest Quarter | mm | * |
| bio19 | Precipitation of Coldest Quarter | mm | * |

Bio Factors represent the bioclimatic variables. Description explains the meaning of each variable. Unit indicates the unit of each variable's data, where "-" denotes no unit. Choose or not indicates whether the variable was selected for subsequent MaxEnt model analysis.

generated under current climatic conditions. Model performance was evaluated using the Boyce index, which assesses the consistency between predicted suitability values and the density of observed occurrences. Positive Boyce index values indicate a good agreement between model predictions and the actual distribution of the species, thereby reflecting the reliability of the model.

## Contribution of climatic variables

To assess the relative importance of each bioclimatic variable in the model predictions, a permutation-based approach was applied. In this method, the values of a single environmental variable were randomly permuted, and the resulting change in the model predictions was compared with the original predictions to quantify the influence of that variable on model output. The importance of each variable was expressed as a percentage of permutation importance, indicating its relative contribution to the predicted habitat suitability.

## Habitat suitability classification and area change analysis

Based on the predicted habitat suitability values at species occurrence points under current climatic conditions, the continuous suitability predictions were classified using the 5% training presence threshold. Areas with suitability values above this threshold were further classified into Low Suitability Areas (LSA), Moderate Suitability Areas (MSA), and High Suitability Areas (HSA) by equally partitioning the range between the threshold and the maximum suitability value (1) into three intervals. The same 5% training presence threshold was applied to both current and future climate scenarios to ensure

comparability across predictions. The proportion of each suitability class was calculated under different climate scenarios, and the corresponding spatial area was estimated. This approach allowed for the assessment of changes in the extent and structure of suitable habitats in response to climate change.

## Results

### MaxEnt model parameter optimization and predictive performance

The parameter tuning results indicated that different combinations of feature classes and regularization multipliers had significant effects on model performance. Based on the AICc criterion, the optimal models for both *R. roxburghii* and *G. molesta* were obtained using the LQH feature combination with a regularization multiplier of 1 (ΔAICc = 0) (Fig 2). Furthermore, evaluation of the predictive performance of the best models showed positive Boyce indices for both species (*R. roxburghii*: 0.975; *G. molesta*: 0.918), indicating a high consistency between predicted habitat suitability and the actual species distributions (Fig 3).

### Climate variable permutation importance analysis

The permutation importance of climatic variables on model predictions is shown in Fig 4. For *R. roxburghii*, bio6 (Minimum Temperature of the Coldest Month) contributed the most to the model, followed by bio2 (Mean Diurnal Range) and bio19 (Precipitation of the Coldest Quarter), with permutation importance values of 28.2%, 23.8%, and 21.6%, respectively. For *G. molesta*, bio6 also exhibited the highest importance, followed by bio15 (Precipitation Seasonality) and bio18 (Precipitation of the Warmest Quarter), with permutation importance values of 64.6%, 14.0%, and 13.3%, respectively.

### Potential suitable habitat of *R. roxburghii* under current and future climate scenarios

MaxEnt model results (Fig 5) indicate that under current climatic conditions (1970–2000), the total suitable area (non-UA) for *R. roxburghii* in China is ~1.4795 million km², accounting for 15.41% of the national territory (S3 Table). Among this, the high suitability area (HSA) covers about 0.4221 million km² (~4.40%), and is primarily concentrated in southwestern China. The moderate suitability area (MSA) totals approximately 0.3015 million km² (~3.14%), also mainly distributed in southwestern China, generally forming a surrounding zone around the HSA, with a smaller portion extending into eastern China. The low suitability area (LSA) covers approximately 0.7559 million km² (~7.87%) and shows a much broader distribution, spanning southwestern, central, and eastern China, with substantially greater spatial coverage than both HSA and MSA.

Under the SSP2–4.5 scenario for 2061–2080, the total potential suitable area of *R. roxburghii* is 1.4834 million km², showing little overall change compared with current climate conditions. The primary difference lies in the proportion of HSA

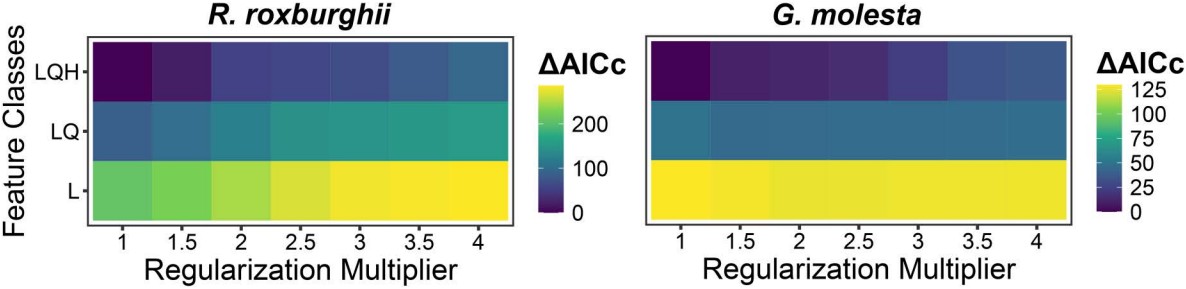

**Fig 2. ΔAICc values of *R. roxburghii* (left) and *G. molesta* (right) under different combinations of feature classes and regularization multipliers.** The x-axis represents the regularization multiplier, tested from 1 to 4 with a step size of 0.5. The y-axis indicates the feature class combinations: linear (L), linear + quadratic (LQ), and linear + quadratic + hinge (LQH).

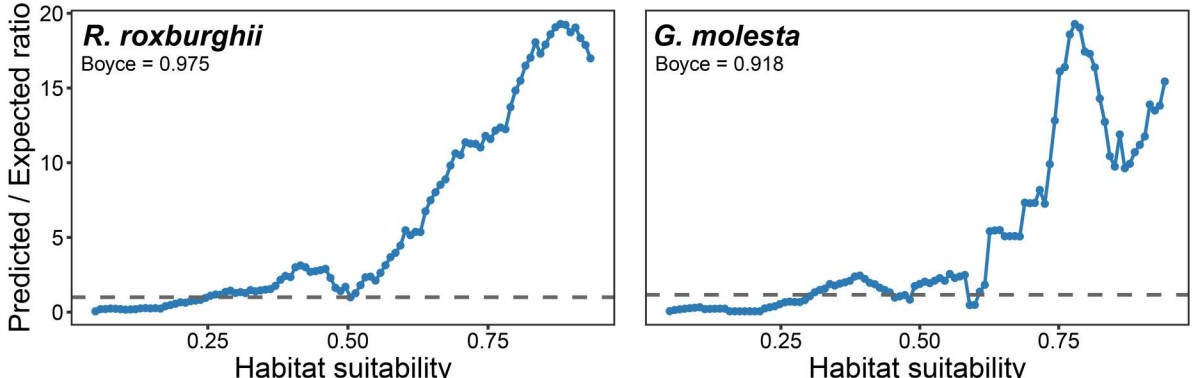

**Fig 3. Model performance evaluation for the optimal models of *R. roxburghii* (left) and *G. molesta* (right).** The x-axis represents the predicted habitat suitability, and the y-axis indicates the ratio of predicted to expected occurrence frequency (Predicted/Expected ratio). The dashed line represents the reference where predicted frequency equals expected frequency. Values above 1 indicate relative enrichment of species occurrence points within the corresponding suitability interval, demonstrating consistency between model predictions and observed distributions. The Boyce index quantifies the correlation between predicted habitat suitability and observed occurrence frequency; positive values indicate that model predictions are better than random, with higher values reflecting superior predictive performance.

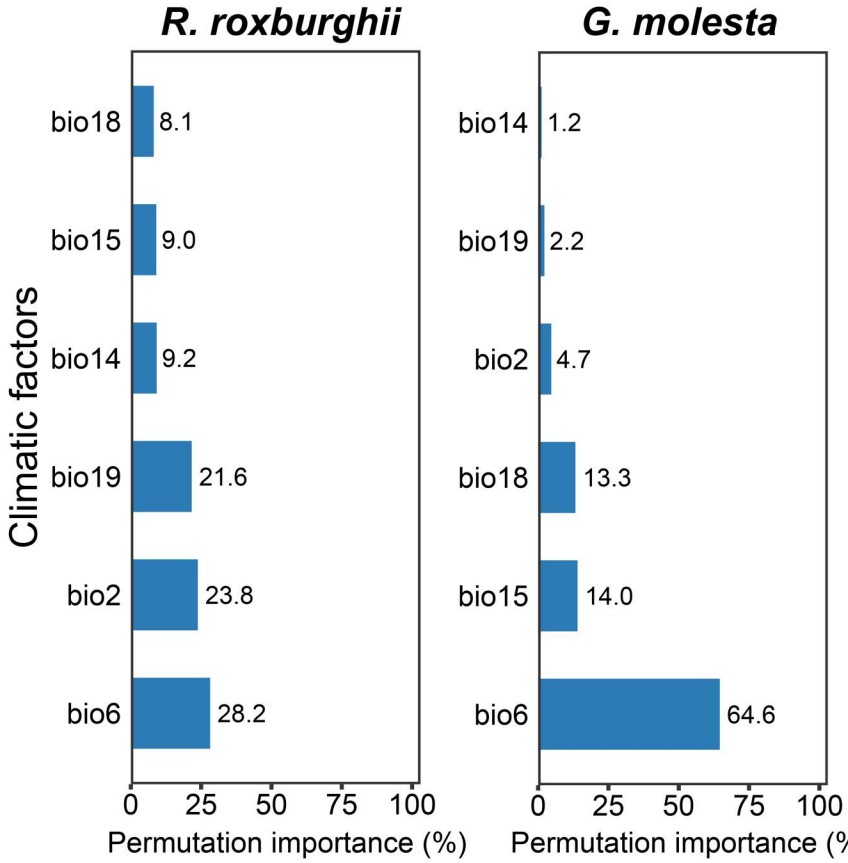

**Fig 4. Permutation importance of selected bioclimatic variables for *R. roxburghii* (left) and *G. molesta* (right).** Higher values indicate greater influence of the variable on model predictions.

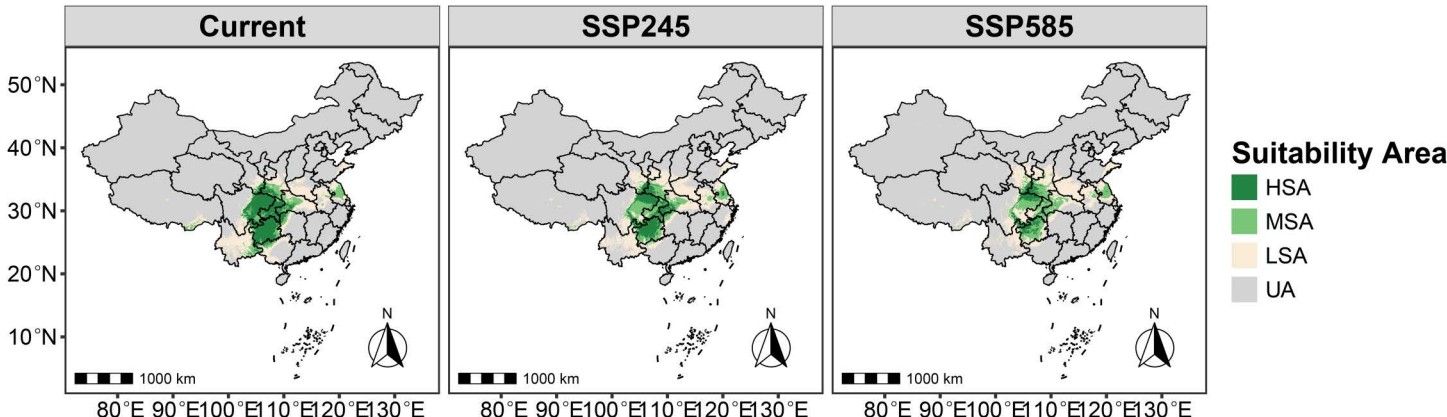

**Fig 5. Potential suitable habitat of *R. roxburghii* under current and future climate scenarios.** Different colors indicate habitat suitability categories, including low, moderate, and high suitability areas. Map Review Number: GS(2019)1822.

and LSA. Compared to the present, HSA decreases to 2.88% of the national territory (~0.2760 million km²), representing a reduction of ~1.5% (~0.144 million km²). This decline is mainly driven by the transition of a core high-suitability region under the current climate, which is located in eastern Sichuan and Chongqing, into moderate suitability areas (MSA). Conversely, LSA increases to 8.82%, representing a ~1% increase in area. Under the SSP5–8.5 high-emission scenario, changes in potential suitable habitat are more pronounced. The total suitable area declines to 1.3941 million km², ~0.1 million km² less than in the previous scenarios. In addition, the proportion of HSA further shrinks to 1.55%, and its distribution becomes increasingly fragmented rather than forming continuous regions. Analysis of suitability proportion maps (Fig 6) indicates that with progressively adverse future climatic conditions, HSA is likely to convert into MSA, resulting in a gradual decrease of HSA and a corresponding increase in MSA, while the proportions of LSA and UA remain relatively stable.

## Potential suitable habitat of *G. molesta* under current and future climate scenarios

Under the current climate scenario, the total suitable area (non-UA) for *G. molesta* is 2.3513 million km² (S4 Table), accounting for 24.49% of China's land area, which is substantially larger than that of *R. roxburghii*. The high suitability area (HSA) covers 0.3940 million km², representing 4.10% of the national territory, and is primarily located in northern China and small portions of southwestern China, particularly northeastern Sichuan Province (Fig 7), comparable in size to the HSA of *R. roxburghii*. The moderate suitability area (MSA) spans 0.3838 million km² (4% of the national territory), mainly surrounding the HSA. The low suitability area (LSA) covers 1.5735 million km² (16.39%), roughly twice the LSA of *R. roxburghii*, encompassing nearly all southern provinces of China.

Under future climate scenarios SSP2–4.5 and SSP5–8.5, the total suitable area of *G. molesta* is projected to increase progressively (Fig 8), reaching 2.9194 million km² (30.41%) and 3.4386 million km² (35.82%), respectively. The most pronounced changes occur in the HSA and MSA. The HSA proportion increases to 8.93% (~0.8569 million km²) and 12.33% (~1.1837 million km²), while the MSA proportion rises to 6.62% (~0.6358 million km²) and 7.81% (~0.7498 million km²), respectively. Notably, with climate change, the proportion of unsuitable areas (UA) for *G. molesta* gradually decreases to 64.18%, whereas the UA proportion for *R. roxburghii* remains nearly unchanged.

## Potential overlapping distribution of *R. roxburghii* and *G. molesta* and its changes under current and future climate scenarios

Under current climate conditions, the total overlapping area of *R. roxburghii* and *G. molesta* (HSA+MSA+LSA) accounts for ~ 1.0368 million km² (~10.8% of China's land area) (Fig 9). This overlapping region is primarily concentrated in

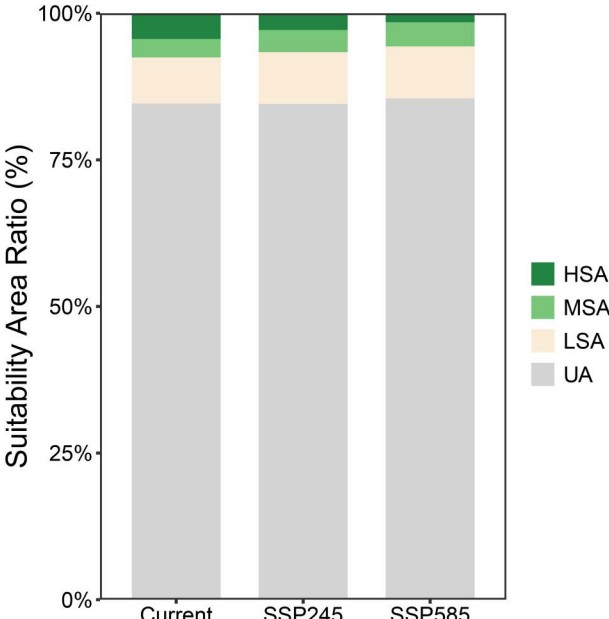

**Fig 6. Stacked bar chart showing the proportional area of *R. roxburghii* habitat suitability under current and future climate scenarios.** The x-axis represents different climate scenarios, and the y-axis represents the proportion of each suitability category. Different colors indicate low, moderate, and high suitability areas.

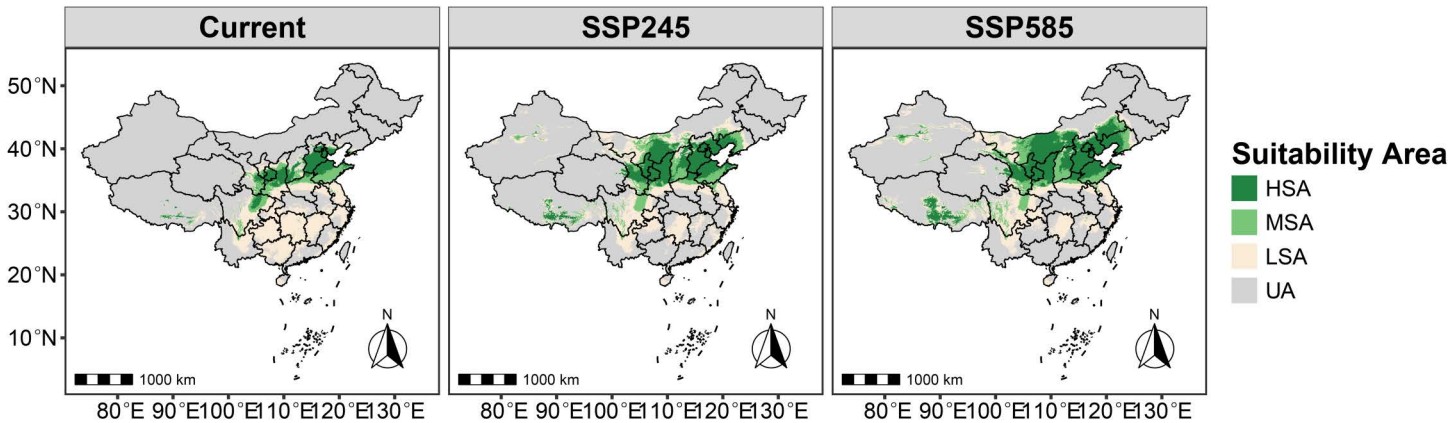

**Fig 7. Potential suitable habitat of *G. molesta* under current and future climate scenarios.** Different colors indicate habitat suitability categories, including low, moderate, and high suitability areas. Map Review Number: GS(2019)1822.

southwestern China, with additional distribution in adjacent transitional zones extending into central and eastern China. Within the overall overlapping area, the regions of MSA and HSA overlap are mainly concentrated in the southwestern core zone of the total overlap. However, the combined overlap of HSA and MSA occupies a relatively small proportion, covering approximately 1.4% of China's land area (~0.1344 million km²) (Fig 10). In particular, the HSA overlap is extremely limited, covering roughly 28,800 km² (~0.3% of China's land area) (Fig 11). For *R. roxburghii*, under current

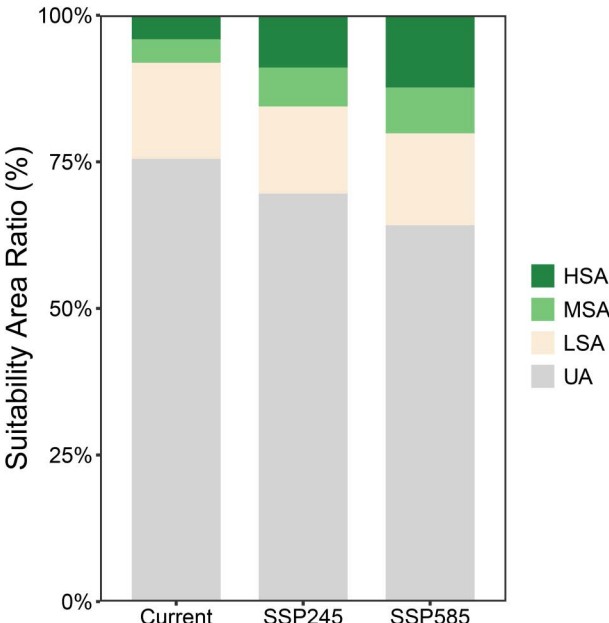

**Fig 8. Stacked bar chart showing the proportional area of *G. molesta* habitat suitability under current and future climate scenarios.** The x-axis represents different climate scenarios, and the y-axis represents the proportion of each suitability category. Different colors indicate low, moderate, and high suitability areas.

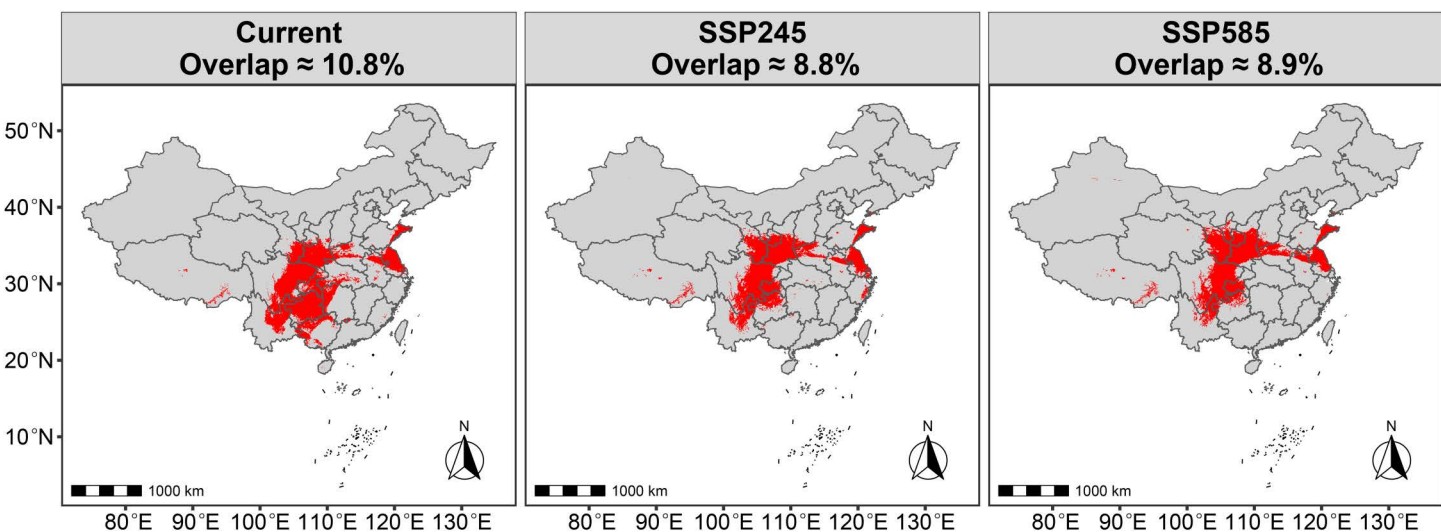

**Fig 9. Distribution of *R. roxburghii* and *G. molesta* suitable areas (HSA+MSA+LSA) under current and future climate scenarios.** Red areas indicate regions where the suitable habitats of the two species overlap. Map Review Number: GS(2019)1822.

climatic conditions, the overall overlap largely coincides with its potential suitable distribution range, indicating that most suitable habitats of *R. roxburghii* may fall within areas where co-occurrence with *G. molesta* is possible.

Under the SSP2–4.5 scenario, the proportion of overlapping suitable area decreases to 8.8%, accompanied by a general northward shift. The overlapping region expands in northwestern and northern transitional zones, while contracting

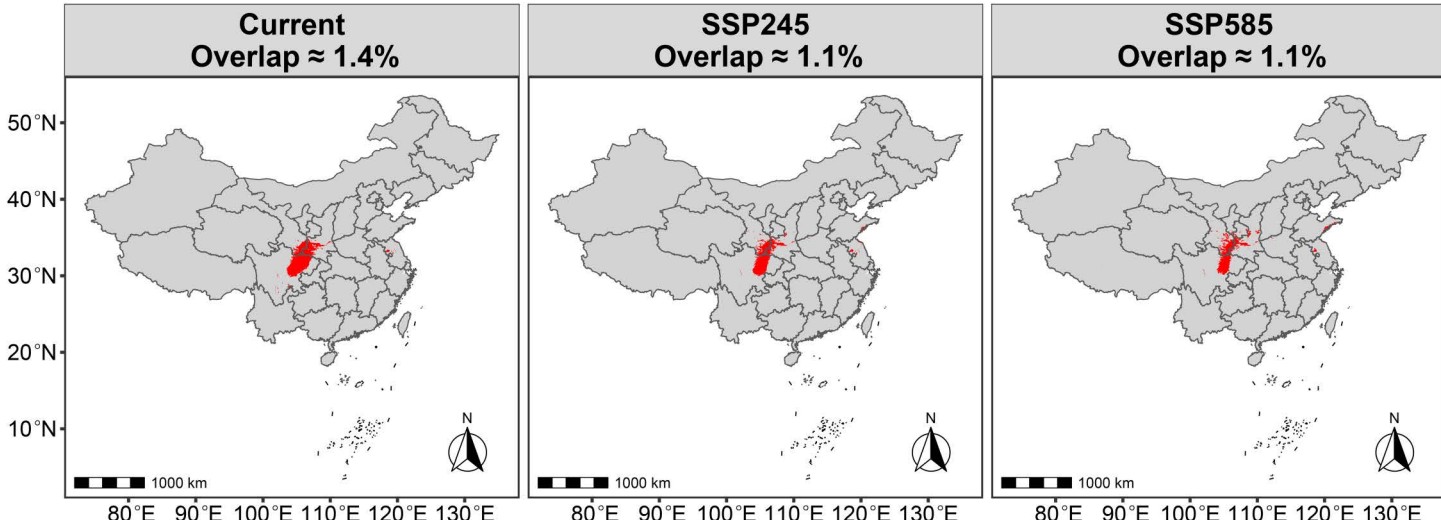

**Fig 10. Overlapping moderate- to high-suitability areas (HSA+MSA) of *R. roxburghii* and *G. molesta* under current and future climate scenarios.** Red areas indicate regions where the suitable habitats of the two species overlap. Map Review Number: GS(2019)1822.

in southwestern China, where suitability is reduced. Under the higher emission SSP5–8.5 scenario, the total proportion of overlapping suitable areas remains relatively stable. However, the northward expansion of the overlap becomes even more pronounced, indicating that while the overall area does not increase significantly, the spatial pattern of potential pest risk is likely to undergo a notable northward redistribution. For example, the northward shift of the overall overlapping area leads to two main patterns: (1) the overlap between the highly suitable areas (HSA) of *R. roxburghii* and *G. molesta* nearly disappears (Fig 11); and (2) *R. roxburghii* retains a relatively isolated suitable area in southwestern China (e.g., Guizhou Province), where the co-occurrence with *G. molesta* is minimal (Figs 5,9).

## Discussion

### Climate factors affecting the distribution of *R. roxburghii* and *G. molesta*

Among the climate variables selected in this study, bio6 (Minimum Temperature of the Coldest Month) exhibited the highest permutation importance, indicating that it plays a dominant role in shaping the potential distributions of both *R. roxburghii* and *G. molesta*. This finding is highly consistent with previous studies demonstrating the limiting effects of temperature on plant growth and insect survival [3,4,27,28]. For *R. roxburghii*, low temperatures constitute a key constraint on its geographic distribution [29]. As a temperate–subtropical perennial fruit tree, its growth and flowering processes depend to some extent on winter low temperatures; however, extreme cold can significantly inhibit flower bud differentiation and increase the risk of frost damage, thereby limiting its expansion into high-latitude and high-altitude regions [30,31]. The high importance of bio6 reflects *R. roxburghii*'s sensitivity to winter minimum temperature thresholds and explains why its potential suitable habitats are primarily concentrated in the warm and humid regions of southwestern and central China. Similarly, bio6 is the most critical environmental variable for *G. molesta*, which is closely related to its overwintering biology [17]. *G. molesta* overwinters as larvae or pupae, and its survival rate during winter largely depends on extreme low-temperature conditions; sustained temperatures below its cold tolerance threshold can significantly reduce overwintering populations, thereby limiting its establishment in colder regions [32,33].

Bio2 (Mean Diurnal Range) showed relatively high importance only in the *R. roxburghii* model, indicating that this species is considerably more sensitive to diurnal temperature fluctuations than *G. molesta*. As a perennial woody fruit tree, *R.*

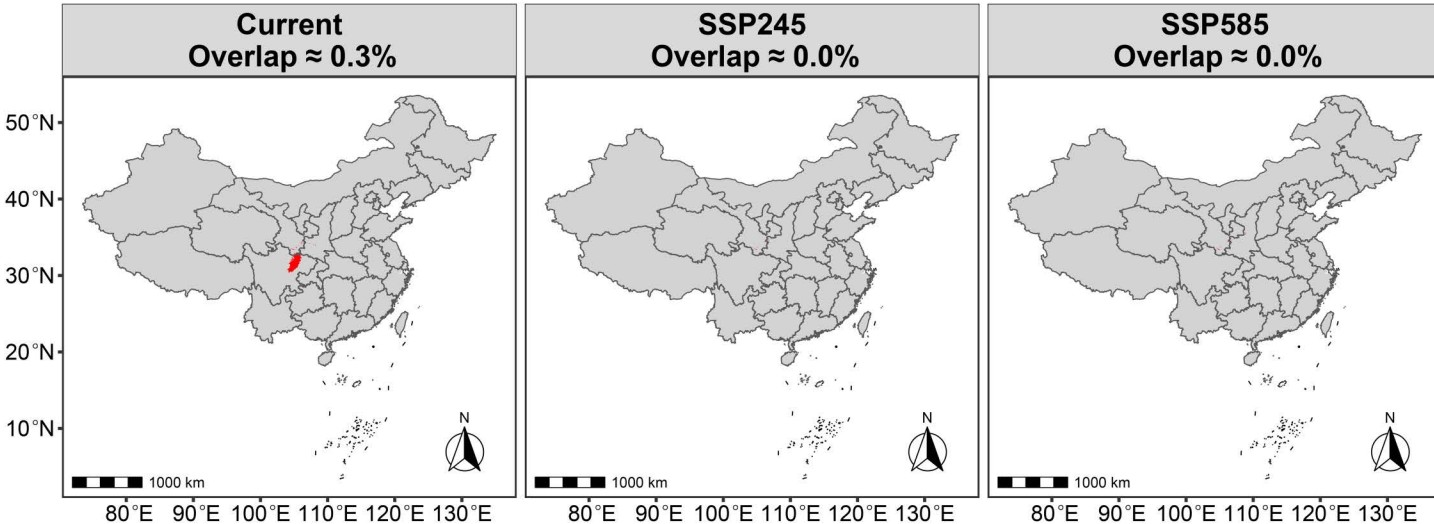

**Fig 11. Overlapping high-suitability areas (HSA) of *R. roxburghii* and *G. molesta* under current and future climate scenarios.** Red areas indicate regions where the suitable habitats of the two species overlap. Map Review Number: GS(2019)1822.

*roxburghii*'s growth, flowering, and fruiting processes rely on relatively stable temperature conditions. Excessive day–night temperature variation can disrupt the balance between photosynthesis and respiration, limiting long-term establishment even in regions with otherwise suitable mean annual temperatures [34]. In contrast, *G. molesta* has some dispersal capacity via flight, which allows it to avoid locally unfavorable conditions over short timescales, reducing its sensitivity to daily temperature fluctuations. Consequently, diurnal temperature range exerts a weaker limiting effect on its distribution [35,36]. Additionally, bio15 (Precipitation Seasonality) and bio18 (Precipitation of the Warmest Quarter) are moderately important for *G. molesta*, likely reflecting the dependency of its larval feeding stages on host plant growth and local moisture conditions [37,38].

Overall, the *G. molesta* model exhibits a clear single-factor dominance pattern, with bio6 accounting for 64.6% of permutation importance, while other climate variables contribute relatively little. This indicates that the potential distribution of *G. molesta* is primarily constrained by the key physiological threshold of extreme winter cold, consistent with its overwintering biology and high cold sensitivity. In contrast, while bio6 remains the most important variable for *R. roxburghii* (28.2%), its distribution is also jointly constrained by multiple climate factors, including bio2 (23.8%) and bio19 (Precipitation of the Coldest Quarter, 21.6%). This multi-factor limitation pattern reflects the species' integrated dependence on temperature stability and moisture conditions, indicating a relatively more complex climatic niche structure for *R. roxburghii* [39–41].

## Current climate distributions and overlap between *R. roxburghii* and *G. molesta*

Under current climatic conditions, the potential suitable habitats of *R. roxburghii* are primarily concentrated in southwestern and central China, with the total suitable area (non-UA) being noticeably smaller than that of *G. molesta* (Figs 5,7). This difference reflects inherent distinctions in ecological niche breadth between the crop and its pest: as a perennial fruit tree, *R. roxburghii* has relatively strict climatic requirements, whereas *G. molesta*, as a highly adaptable agricultural pest, exhibits a much broader potential distribution [6]. Based on the analysis of overall suitable area overlap, most potential suitable habitats of *R. roxburghii* may fall within areas where *G. molesta* could potentially occur. It should be noted that the term "potential threat" in this context refers to the possibility of spatial co-occurrence between the two species, rather than direct pest impact. Most of these overlapping areas correspond to low-suitability (LSA) regions, where the probability of co-occurrence is relatively low under current climatic conditions. Nevertheless, such spatial overlap suggests that

potential pest risk cannot be completely excluded. In addition, the overlap in highly suitable habitats (HSA) between the two species under current climate conditions is very limited, covering only about 0.3% of China's land area (Fig 11). This indicates that, under the present climate, *G. molesta* does not pose a widespread and continuous high-risk threat across all potential habitats of *R. roxburghii*. Instead, the potential risk is spatially restricted to a limited number of regions where environmental conditions are jointly favorable, suggesting a pattern of localized co-occurrence rather than nationwide exposure [42–44]. These limited high-suitability overlap areas are mainly distributed in northeastern Sichuan and its surrounding regions, likely due to the concurrent presence of favorable winter temperatures and precipitation conditions [45]. In addition, anthropogenic factors may also contribute to this pattern, as activities such as fruit trade and the expansion of cultivation can facilitate pest dispersal and increase the likelihood of local establishment [46].

### Future climate distributions and overlap between *R. roxburghii* and *G. molesta*

Under future climate scenarios, the potential distributions of *R. roxburghii* and *G. molesta* exhibit markedly different response patterns. With increasing greenhouse gas emissions, the area of highly suitable habitat (HSA) for *R. roxburghii* continues to contract and gradually becomes fragmented (Figs 5,6). This trend is particularly pronounced under the SSP5–8.5 scenario, indicating that extreme warming conditions may reduce the suitability of traditional core cultivation regions for *R. roxburghii*. This observation aligns with previous studies suggesting that climate warming can decrease habitat suitability for some subtropical fruit trees [22,47–50]. In contrast, *G. molesta* shows a stronger positive response to warming, with both its total suitable area and medium-to-high suitability areas expanding significantly under both future scenarios (Figs 7,8). Notably, under SSP5–8.5, its potential distribution extends northward and into higher-latitude regions. This pattern is consistent with the commonly observed poleward expansion of insect distributions under rising temperatures and reflects the pest's strong environmental adaptability [38,51–53]. Nevertheless, the overlap in medium-to-high and high suitability areas between *R. roxburghii* and *G. molesta* does not increase under future scenarios; instead, it gradually declines (Figs 10,11). This suggests that future climate change will not simply amplify potential pest risk uniformly across the entire distribution of *R. roxburghii*, but is more likely to lead to a spatial reconfiguration of risk, concentrating high-risk areas in regions that maintain critical climatic thresholds.

### Conservation and management strategies

Based on the above findings, pest and disease management for the *R. roxburghii* industry in the future should shift from "broad-scale control" to "targeted management in high-priority areas." Given that the overlap of highly suitable habitats (HSA) between *R. roxburghii* and *G. molesta* remains limited under both current and future climate scenarios, these areas can be considered potential pest risk hotspots and should be prioritized in long-term monitoring and early-warning systems [54,55]. Under current climate conditions, it is recommended to strengthen field surveys and overwintering population monitoring in high-risk overlapping regions, such as northeastern Sichuan and surrounding areas, to mitigate the risk of sudden pest outbreaks. Under future climate scenarios, as the potential distribution of *G. molesta* shifts northward, newly expanded or potential cultivation areas for *R. roxburghii* should also be incorporated into pest risk assessment frameworks to prevent emerging infestations due to newly suitable climatic conditions. Furthermore, management strategies should consider regional climate change trends to guide cultivar selection, planting layout, and timing of control measures. For instance, under warming scenarios, choosing heat-tolerant or low-chilling-requirement *R. roxburghii* varieties, combined with biological control and ecological regulation, could enhance the overall stability and resilience of the production system.

## Conclusions

This study aimed to assess the potential distributions of *R. roxburghii* and its primary pest, *G. molesta*, and to evaluate their spatial overlap under current and future climate scenarios using an optimized MaxEnt model.

The results reveal three key findings. First, *R. roxburghii* exhibits a relatively restricted suitable range compared with *G. molesta*, reflecting its narrower climatic niche. Second, although the total spatial overlap between the two species is relatively extensive under current climate conditions, most of this overlap occurs in low-suitability areas, indicating a generally low probability of co-occurrence. In contrast, overlap within moderate- to high-suitability zones is much more limited and geographically constrained, suggesting that potential interaction zones are primarily localized rather than widespread. Third, under future climate scenarios, *R. roxburghii* shows contraction of highly suitable habitats, whereas *G. molesta* expands northward; however, their overlap does not increase accordingly, but instead shows a spatial redistribution.

These findings suggest that climate change may not uniformly intensify potential pest risk across the host's entire distribution, but rather reshape its spatial pattern by altering the distribution of potential interaction zones. This provides a scientific basis for targeted monitoring and region-specific pest management strategies for *R. roxburghii*.

## Supporting information

**S1 Table. List of literature related to the distribution of *Grapholita molesta*.**
(XLSX)

**S2 Table. The final occurrence records of *Rosa roxburghii* and *Grapholita molesta* used in this study.**
(CSV)

**S3 Table. Proportions of different suitability classes for *Rosa roxburghii* under current and future climate scenarios.**
(XLS)

**S4 Table. Proportions of different suitability classes for *Grapholita molesta* under current and future climate scenarios.**
(XLS)

## Author contributions

**Conceptualization:** Hongling Qin.

**Data curation:** Guoqing Luo.

**Formal analysis:** Guoqing Luo.

**Investigation:** Meifang Lu, Heng Xiang, Zhengbin Li.

**Methodology:** Hongling Qin, Shuangshuang Wu.

**Validation:** Heng Xiang, Zhengbin Li.

**Visualization:** Hongling Qin, Meifang Lu.

**Writing – original draft:** Hongling Qin.

**Writing – review & editing:** Hongling Qin, Shuangshuang Wu.

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
