## [Decision Letter · Decision Letter 0]

17 Dec 2025

Dear Dr. Qin,

Thank you for submitting your manuscript to PLOS ONE. After careful consideration, we feel that it has merit but does not fully meet PLOS ONE’s publication criteria as it currently stands. Therefore, we invite you to submit a revised version of the manuscript that addresses the points raised during the review process.

We look forward to receiving your revised manuscript.

Kind regards,

Umesh Sharma

Academic Editor

PLOS One

Journal Requirements:

3. Please note that PLOS One has specific guidelines on code sharing for submissions in which author-generated code underpins the findings in the manuscript. In these cases, we expect all author-generated code to be made available without restrictions upon publication of the work. Please review our guidelines at https://journals.plos.org/plosone/s/materials-and-software-sharing#loc-sharing-code and ensure that your code is shared in a way that follows best practice and facilitates reproducibility and reuse.

“This research was funded by the 2024 Scientific Research Foundation of Guiyang Institute of Humanities and Technology (Grant No. 2024rwscjs03).”

6. We note that Figure 1, 7, 8, 9, 10, 11, 12 ,13, 14, 15 and 16 in your submission contain [map/satellite] images which may be copyrighted. All PLOS content is published under the Creative Commons Attribution License (CC BY 4.0), which means that the manuscript, images, and Supporting Information files will be freely available online, and any third party is permitted to access, download, copy, distribute, and use these materials in any way, even commercially, with proper attribution. For these reasons, we cannot publish previously copyrighted maps or satellite images created using proprietary data, such as Google software (Google Maps, Street View, and Earth). For more information, see our copyright guidelines: http://journals.plos.org/plosone/s/licenses-and-copyright.

1. You may seek permission from the original copyright holder of Figure 1, 7, 8, 9, 10, 11, 12 ,13, 14, 15 and 16 to publish the content specifically under the CC BY 4.0 license.

7. Please update your submission to use the PLOS LaTeX template. The template and more information on our requirements for LaTeX submissions can be found at http://journals.plos.org/plosone/s/latex.

Reviewers' comments:

Reviewer's Responses to Questions

**Comments to the Author**

1. Is the manuscript technically sound, and do the data support the conclusions?

Reviewer #1: Partly

Reviewer #2: Yes

Reviewer #3: Yes

2. Has the statistical analysis been performed appropriately and rigorously?

Reviewer #1: Yes

Reviewer #2: Yes

Reviewer #3: Yes

3. Have the authors made all data underlying the findings in their manuscript fully available?

Reviewer #1: Yes

Reviewer #2: Yes

Reviewer #3: Yes

4. Is the manuscript presented in an intelligible fashion and written in standard English?

Reviewer #1: Yes

Reviewer #2: Yes

Reviewer #3: No

Reviewer #1: A primary concern is the insufficient optimization of the MaxEnt model parameters. The use of default parameters may lead to excessive model complexity and overfitting. It is strongly recommended to employ R packages like ENMeval to optimize key parameters such as the regularization multiplier and feature classes. The goal should be to select the most parsimonious model, for instance, the one with the lowest delta AICc, and this optimization process should be clearly described in the manuscript. On the topic of environmental variable selection, the current approach is oversimplified. Relying solely on Pearson correlation for variable screening may overlook ecological significance and fails to effectively address multicollinearity. It is suggested to supplement the correlation analysis with techniques like Principal Component Analysis or Variance Inflation Factor analysis. Furthermore, consulting existing literature will help ensure the selected variables have a clear ecological explanation for the species' distributions.

Other minor comments:

1) For model evaluation and validation, the manuscript currently overrelies on a single evaluation metric, the AUC value. While AUC is useful for measuring discriminatory ability, it is not sensitive to the calibration and reliability of predictions. To provide a more robust assessment, it is advisable to include additional metrics like the continuous Boyce index, which measures the consistency between prediction probabilities and observed distributions. Additionally, conducting spatial cross-validation would better test the model's transferability.

2)In the results analysis and discussion section, there is an insufficient interpretation of the key environmental factor bio6, which is the Min Temperature of Coldest Month. The paper identifies it as a core factor but does not deeply explain its specific biological impact. The discussion should be enhanced by integrating the biological characteristics of both species, such as the low-temperature vernalization requirement for R. roxburghii and the overwintering survival rate of G. molesta, to elucidate the critical role of bio6 and add significant depth to the analysis.

3)Concerning the presentation of results, the use of a single future climate change scenario, SSP5-8.5, limits the generalizability of the conclusions. If feasible, it is recommended to incorporate simulations for one to two different emission pathways, such as SSP1-2.6 or SSP2-4.5. Analyzing the uncertainty of distribution changes across different scenarios would make the study more comprehensive and its findings more robust.

4) For language and formatting, some minor errors were noted, such as repeated periods in the abstract and main text. A thorough round of language polishing is recommended before submission to ensure professionalism and fluency. Simultaneously, please strictly check and adhere to the PLoS ONE journal's specific formatting requirements for submissions.

Reviewer #2: The manuscript addresses an applied and relevant topic by using MaxEnt to predict the current and future potentially suitable areas for Rosa roxburghii and its major pest Grapholita molesta, as well as their spatial overlap. The study has clear implications for industrial planning and pest‐risk early warning. The paper follows a conventional structure, and the overall analytical workflow is generally coherent. Nevertheless, several issues require further revision before the manuscript can be considered for publication. They are as follows:

1. Insufficient clarity on the literature data.

The manuscript does not specify the temporal scope of the literature survey (i.e., which years were covered). In addition, the number of valid/eligible publications included after screening should be explicitly reported. This information is essential for evaluating data completeness and potential bias.

2. Paragraph duplication in “Species Distribution Data.”

In the Materials and Methods section (Species Distribution Data), the second and third paragraphs are nearly identical. Both repeat the same six filtering criteria for occurrence records with only minor wording differences. This appears to be a clear editing or copy–paste error. Please remove one of the duplicated paragraphs and carefully check the manuscript to ensure no similar repetitions remain elsewhere.

3. Unclear map-related statement.

Please clarify whether the manuscript should report a map survey number or a map approval number for China map.

4. Overly generic protection strategies.

The section on Protection Strategies remains too broad and descriptive. The authors should provide more concrete, feasible, and actionable recommendations (e.g., targeted monitoring in projected overlap hotspots, region-specific management timelines, or practical cultivation/pest-control adjustments under future climate scenarios). Strengthening this part would improve the applied value of the study.

Overall, the study is promising, but addressing the above points will substantially enhance its rigor, transparency, and practical relevance.

Reviewer #3: This paper predicts the current and future potential distribution of Rosa roxburghii and its pest Grapholita molesta under climate change, and identifies their overlapping high-suitability areas to assess potential pest risk and guide regional management. Overall, the study is methodologically sound. The manuscript has merit and can make a useful contribution after revision. The main issues that need to be addressed relate to (1) imprecise use of ecological terminology, (2) an underdeveloped introduction that lacks sufficient background and references to previous studies, and (3) insufficient justification for environmental variable selection. Clarifying these points will significantly improve the scientific rigor and readability of the manuscript.

Specific Comments

1. Terminology: The term “predation” is used to describe the feeding behavior of G. molesta on R. roxburghii. In ecological terminology, predation refers specifically to animals killing and consuming other animals. In this case, the correct term is “herbivory.” The terminology should be corrected throughout the manuscript to avoid conceptual confusion.

2. Background on Grapholita molesta: The introduction lacks sufficient background on G. molesta, a pest that has been extensively studied, particularly in China. The authors are encouraged to expand the introduction to include: Known distribution patterns of G. molesta；Typical dispersal pathways (including the role of trade and human-mediated movement)； Differences between infested and non-infested regions；Its host range, as G. molesta attacks multiple fruit crops and is not restricted to R. roxburghii. Providing this context is important for interpreting the ecological and management implications of the distribution overlap results.

3. The selection criteria for environmental variables for both the pest and the host plant appear nearly identical and could be presented together to reduce redundancy.

4. The manuscript states that variables with correlation coefficients greater than 0.9 were excluded. This threshold is relatively permissive, and highly correlated variables may still enter the model. The authors should: 1) provide references supporting this threshold, or 2) further justify their choice of variables based on ecological knowledge and previous modeling studies. 3) relevant literature on variable selection in distribution modeling for R. roxburghii and G. molesta should be cited.

5. The phrase “The survey map number” appears to be incorrect and should be revised to “Map Review Number.”

6. Under current climate conditions, the overlap between high suitability areas (HSA) of R. roxburghii and G. molesta is very limited. This suggests that G. molesta may not pose a widespread threat across the entire suitable range of R. roxburghii.

7. Figures 12 and 13 appear redundant; Figure 13 alone would be sufficient.

8. The authors may consider merging distribution maps into composite panels (e.g., current vs. future scenarios) to reduce repetition and improve visual clarity.

9. Discussion Clarity: Some statements in the discussion are potentially confusing. For example, the manuscript states that the “HSA of R. roxburghii is similar in size to that of G. molesta,” while earlier results indicate large differences in total suitable area. If this comparison refers specifically to high suitability areas, this should be stated explicitly to avoid misunderstanding.

10. The discussion of limited HSA overlap could be strengthened by emphasizing its ecological and management implications. Given that the overlapping HSA represents only 0.33% of China’s land area, these regions may represent priority hotspots for monitoring and targeted pest management.

.

Reviewer #1: No

Reviewer #2: No

Reviewer #3: No

---

## [Author Response · Author response to Decision Letter 1]

7 Mar 2026

We sincerely thank the Editor and the anonymous reviewers for their constructive and insightful comments, which have greatly helped us to improve the quality and clarity of the manuscript. All comments have been carefully considered and addressed point by point below.

Reviewer #1

Comment 1:

A primary concern is the insufficient optimization of the MaxEnt model parameters. The use of default parameters may lead to excessive model complexity and overfitting. It is strongly recommended to employ R packages like ENMeval to optimize key parameters such as the regularization multiplier and feature classes. The goal should be to select the most parsimonious model, for instance, the one with the lowest delta AICc, and this optimization process should be clearly described in the manuscript. On the topic of environmental variable selection, the current approach is oversimplified. Relying solely on Pearson correlation for variable screening may overlook ecological significance and fails to effectively address multicollinearity. It is suggested to supplement the correlation analysis with techniques like Principal Component Analysis or Variance Inflation Factor analysis. Furthermore, consulting existing literature will help ensure the selected variables have a clear ecological explanation for the species' distributions.

Response to Comment 1:

We appreciate the reviewer’s detailed and constructive comments on the model construction methodology. We fully agree that parameter optimization and environmental variable selection in MaxEnt modeling are critical for reducing overfitting and improving predictive reliability, and we have revised the manuscript accordingly to clarify these points.

(1) Optimization of MaxEnt model parameters

In response to the reviewer’s suggestions regarding parameter optimization, we systematically tuned the key parameters of the MaxEnt models using the ENMeval package. Specifically, MaxEnt modeling was implemented in the R environment based on the maxnet algorithm, and a comprehensive set of candidate models was evaluated by testing different feature class combinations (L, LQ, and LQH) and regularization multiplier values (1–4, with an interval of 0.5). During model evaluation, spatial block cross-validation was applied to reduce the potential influence of spatial autocorrelation on model performance. The small-sample corrected Akaike Information Criterion (AICc) was used as the model selection criterion, and the model with ΔAICc = 0 was selected as the optimal model. As a result, the optimal models for both species were characterized by the LQH feature class combination with a regularization multiplier of 1. These results are now clearly described in the revised manuscript. In addition, the predictive performance of the optimal models was further evaluated using the continuous Boyce index (Fig 3). The Boyce indices for both species were significantly positive, indicating a strong agreement between predicted habitat suitability and observed species occurrences. All of these methodological details and results have been added to the Methods and Results sections of the revised manuscript.

(2) Environmental variable selection

We agree that relying solely on Pearson correlation coefficients may be insufficient to fully address multicollinearity among environmental variables. Therefore, in the revised manuscript, we explicitly clarify that a two-step variable selection strategy was adopted, integrating correlation analysis, multicollinearity diagnostics, and ecological interpretability. First, Pearson correlation coefficients were calculated among the 19 bioclimatic variables, and variable pairs with |r| > 0.7 were considered highly correlated. Based on this screening, variables with well-established physiological or ecological relevance in previous studies—such as the minimum temperature of the coldest month (bio6)—were preferentially retained, while other highly correlated variables were removed. Second, variance inflation factor (VIF) analysis was applied to the remaining variables to further diagnose multicollinearity. Only variables with VIF values < 5 were retained for model construction. Compared with the previous screening threshold of |r| > 0.9, this revised procedure imposes a more stringent control on multicollinearity while ensuring the retention of key ecologically meaningful predictors. This revised variable selection workflow is now described more clearly in the manuscript.

(3) Ecological interpretation and literature support for environmental variables

Regarding the ecological justification of variable selection, we have further strengthened the biological interpretation of key variables (particularly bio6, bio2, and precipitation-related variables) in the Discussion section. These interpretations are supported by relevant published studies and explain the intrinsic relationships between these variables, the growth characteristics of R. roxburghii, and the overwintering biology of G. molesta. The corresponding explanations and references have been added and refined in the revised manuscript. In summary, we believe that the revised model parameter optimization procedure and environmental variable selection approach are methodologically robust and ecologically well grounded. We also sincerely thank the reviewer for their valuable suggestions, which have helped to further improve the methodological transparency and scientific rigor of the manuscript.

Comment 2:

For model evaluation and validation, the manuscript currently overrelies on a single evaluation metric, the AUC value. While AUC is useful for measuring discriminatory ability, it is not sensitive to the calibration and reliability of predictions. To provide a more robust assessment, it is advisable to include additional metrics like the continuous Boyce index, which measures the consistency between prediction probabilities and observed distributions. Additionally, conducting spatial cross-validation would better test the model's transferability.

Response to Comment 2:

We thank the reviewer for the constructive suggestions regarding model evaluation. We agree that relying solely on the AUC metric is insufficient to comprehensively assess the reliability and calibration performance of model predictions. In the revised manuscript, we therefore incorporated the continuous Boyce index as a complementary evaluation metric to assess the consistency between predicted suitability values and observed species occurrences. The Boyce index is particularly appropriate for presence–background modeling approaches, such as MaxEnt, and provides a more informative assessment of model reliability. In addition, during model calibration and validation, we applied a spatial block cross-validation approach to reduce the potential influence of spatial autocorrelation on model evaluation and to more effectively assess the spatial transferability of the models. The corresponding methods and results have been added and described in detail in the revised Methods and Results sections.

Comment 3:

In the results analysis and discussion section, there is an insufficient interpretation of the key environmental factor bio6, which is the Min Temperature of Coldest Month. The paper identifies it as a core factor but does not deeply explain its specific biological impact. The discussion should be enhanced by integrating the biological characteristics of both species, such as the low-temperature vernalization requirement for R. roxburghii and the overwintering survival rate of G. molesta, to elucidate the critical role of bio6 and add significant depth to the analysis.

Response to Comment 3:

We thank the reviewer for pointing out the insufficient ecological interpretation of the key environmental variable bio6 (minimum temperature of the coldest month). We agree that this aspect of the discussion required further elaboration. In the revised manuscript, we have substantially strengthened the discussion of the biological significance of bio6, integrating the ecological and physiological characteristics of both study species. Specifically, for R. roxburghii, we expanded the discussion on its sensitivity to winter low temperatures as a perennial woody fruit tree, emphasizing the constraining effects of low temperature on flower bud differentiation and its geographic distribution. For G. molesta, we further clarified the importance of extreme low-temperature conditions for overwintering survival and population persistence, considering its biological trait of overwintering in larval or pupal stages. By incorporating these ecological and physiological mechanisms, the limiting role of bio6 as a core environmental factor has been more thoroughly and explicitly explained in the revised Discussion section.

Comment 4:

Concerning the presentation of results, the use of a single future climate change scenario, SSP5-8.5, limits the generalizability of the conclusions. If feasible, it is recommended to incorporate simulations for one to two different emission pathways, such as SSP1-2.6 or SSP2-4.5. Analyzing the uncertainty of distribution changes across different scenarios would make the study more comprehensive and its findings more robust.

Response to Comment 4:

The reviewer raised an important point regarding the selection of future climate scenarios. Relying on a single high-emission scenario may limit the generality of the conclusions. In the revised manuscript, the future projections were therefore expanded by adding the SSP2-4.5 intermediate-emission scenario in addition to the original SSP5-8.5 scenario. A comparative analysis of the projected changes in potential suitable habitats for R. roxburghii and G. molesta was conducted under both scenarios. By contrasting results across multiple emission pathways, the uncertainty associated with species distribution responses to different climate trajectories was more comprehensively evaluated, thereby improving the robustness and applicability of the study conclusions. The relevant results and interpretations have been incorporated into the revised Results and Discussion sections.

Comment 5:

For language and formatting, some minor errors were noted, such as repeated periods in the abstract and main text. A thorough round of language polishing is recommended before submission to ensure professionalism and fluency. Simultaneously, please strictly check and adhere to the PLoS ONE journal's specific formatting requirements for submissions.

Response to Comment 5:

We appreciate the reviewer’s comments regarding language and formatting. In response, the entire manuscript was carefully revised for language clarity and consistency, and minor issues such as repeated punctuation in the abstract and main text were corrected. In addition, the manuscript was thoroughly checked and adjusted in strict accordance with the PLOS ONE submission guidelines, including the structure, figures, tables, references, and overall formatting, to ensure professionalism and compliance with journal standards.

Reviewer #2

Comment 1:

Insufficient clarity on the literature data.

The manuscript does not specify the temporal scope of the literature survey (i.e., which years were covered). In addition, the number of valid/eligible publications included after screening should be explicitly reported. This information is essential for evaluating data completeness and potential bias.

Response to Comment 1:

We appreciate the reviewer’s valuable comments. In this study, the systematic literature survey was conducted primarily for G. molesta, as the occurrence records of R. roxburghii available in public databases were already sufficiently abundant to meet the analytical requirements of this study, and therefore an additional systematic literature search for R. roxburghii was not performed. For G. molesta, literature searches were carried out in both the Web of Science and China National Knowledge Infrastructure (CNKI) databases. The temporal scope of the search was restricted to publications from 1970 to the present, consistent with the baseline period (1970–2000) of the climate data used in this study. The initial search yielded a total of 2,772 publications (786 from Web of Science and 1,986 from CNKI). These records were then screened on a study-by-study basis according to clearly defined inclusion and exclusion criteria, resulting in 36 publications that contained explicit records of the natural distribution of G. molesta and were retained for subsequent analyses. All of the above information has been clearly described in the revised Methods section and summarized in the Supporting Information (S1 Table).

Comment 2:

Paragraph duplication in “Species Distribution Data.”

In the Materials and Methods section (Species Distribution Data), the second and third paragraphs are nearly identical. Both repeat the same six filtering criteria for occurrence records with only minor wording differences. This appears to be a clear editing or copy–paste error. Please remove one of the duplicated paragraphs and carefully check the manuscript to ensure no similar repetitions remain elsewhere.

Response to Comment 2:

We are grateful to the reviewer for identifying the paragraph duplication in the Species Distribution Data section. Upon careful review, the second and third paragraphs were indeed found to be highly repetitive due to an editorial oversight. In the revised manuscript, the redundant paragraph has been removed, and a single, consolidated description of the distribution data screening criteria has been retained. In addition, the entire manuscript was systematically checked to ensure that no similar duplication or formatting errors remain. The reviewer’s attention to this detail is sincerely acknowledged.

Comment 3:

Unclear map-related statement.

Please clarify whether the manuscript should report a map survey number or a map approval number for China map.

Response to Comment 3:

The reviewer raised a concern regarding the source and compliance of the map base layers used in this study. The China base map adopted in the manuscript was obtained from the National Platform for Common GeoSpatial Information Service of China, which provides officially released standard maps compiled in accordance with national and international boundary delineation standards. These standard maps are permitted for public uses such as academic publications, book illustrations, and the presentation of scientific results, and are freely accessible to the public. According to the platform’s usage regulations, the Map Review Number must be indicated when standard maps are directly used. Accordingly, the corresponding Map Review Numbers have now been clearly labeled in the revised figures (e.g., Fig 1), and this information has been added to the figure captions to improve transparency and ensure compliance with map usage requirements.

Comment 4:

Overly generic protection strategies.

The section on Protection Strategies remains too broad and descriptive. The authors should provide more concrete, feasible, and actionable recommendations (e.g., targeted monitoring in projected overlap hotspots, region-specific management timelines, or practical cultivation/pest-control adjustments under future climate scenarios). Strengthening this part would improve the applied value of the study.

Response to Comment 4:

Thank you for the reviewer’s valuable comments. We agree that the “Protection Strategies” section in the original manuscript was relatively general in its description, and that its practical specificity required further improvement. Following the reviewer’s suggestions, we have revised and refined this section to make the management recommendations more concrete, feasible, and directly linked to the model results. After revision, the protection strategies are explicitly based on the overlap analysis of suitable habitats between R. roxburghii and G. molesta, and the following specific measures are proposed: (1) high-suitability overlap hotspots identified under current climate conditions (e.g., northeastern Sichuan and surrounding areas) are designated as priority monitoring regions, where intensified field surveys and overwintering population monitoring are recommended and in

---

## [Decision Letter · Decision Letter 1]

23 Mar 2026

Dear Dr. Qin,

We look forward to receiving your revised manuscript.

Kind regards,

Umesh Sharma

Academic Editor

PLOS One

**Journal Requirements:**

Reviewers' comments:

Reviewer's Responses to Questions

**Comments to the Author**

Reviewer #3: All comments have been addressed

2. Is the manuscript technically sound, and do the data support the conclusions?

Reviewer #3: Yes

3. Has the statistical analysis been performed appropriately and rigorously?

Reviewer #3: Yes

4. Have the authors made all data underlying the findings in their manuscript fully available?

Reviewer #3: Yes

5. Is the manuscript presented in an intelligible fashion and written in standard English?

Reviewer #3: Yes

**Reviewer #3:** I appreciate the efforts of the authors in improving the quality of the manuscript; however, several issues still need to be addressed. I appreciate the efforts of the authors in improving the quality of the manuscript; however, several issues still need to be addressed. I appreciate the efforts of the authors in improving the quality of the manuscript; however, several issues still need to be addressed. I appreciate the efforts of the authors in improving the quality of the manuscript; however, several issues still need to be addressed.

Line 24: Expressing the distribution as “0.3% of the country” is not very informative. If the intention is to emphasize the limited distribution of this species, it would be clearer to report the absolute area. Given that this species is mainly restricted to regions of southwest China, providing exact area values would improve clarity.

Line 43-45: The meaning is somewhat unclear. The term “overlapping ecological niches” is not appropriate here and could be replaced with “co-occurrence”. The statement also conflates climatic effects with niche overlap, as pest–crop interactions already imply spatial co-occurrence; climate primarily modulates their distribution and interaction intensity rather than determining overlap per se.

Line 52: The use of “globally” is inappropriate here; “worldwide” or “across the world” would be more appropriate

Line 134-139: Please clarify the source of the data. If a website was used, the full URL and access date should be provided.

Line 204-207: "areas above the threshold were further divided into Low Suitability Areas (LSA), Moderate Suitability Areas (MSA), and High Suitability Areas (HSA). " Please clarify how this classification was performed. Were these categories defined using equal intervals or another thresholding method?

Line 214-247: The description of the current potential distribution is overly detailed and somewhat difficult to follow. Listing numerous province names does not necessarily improve clarity, especially for an international readership unfamiliar with China’s administrative regions. I suggest simplifying the presentation by focusing on key patterns rather than exhaustive geographic enumeration. For example, the authors could (1) report the total suitable area, (2) summarize the spatial pattern using broader geographic descriptors (e.g., southwestern, central, or eastern China), and (3) highlight where high-suitability areas are concentrated along with their corresponding area values. This more structured and synthetic description would make the results clearer and more informative.

Line 296-310: The use of “percentage of China’s land area” is not very informative for assessing ecological or agricultural significance. It would be more meaningful to report absolute area values and/or the proportion of overlap relative to the total suitable area of the host plant. In addition, the comparison based on high-suitability overlap (HSA) is potentially misleading. Pest damage can occur wherever the pest and host co-occur, regardless of suitability class, and areas with moderate suitability may still support populations capable of causing substantial yield loss. I suggest simplifying the analysis by focusing first on the overall spatial overlap between pest and host suitable areas, and then evaluating how this overlap is distributed across the host’s suitability gradient. This would provide a clearer and more ecologically meaningful assessment of potential risk.

Line367-371, 411: The interpretation appears to assume that spatial overlap directly reflects pest impact, which is not necessarily valid. As noted above, overlap does not equal impact strength, and this distinction should be more clearly addressed.

The conclusion is overly long and reads more like a condensed repetition of the Results and Discussion sections. It would benefit from a more concise and focused synthesis of the key findings. I suggest restructuring this section to clearly highlight (1) the main objective, (2) two to three key findings, and (3) the primary implication of the study.

.

Reviewer #3: No

---

## [Author Response · Author response to Decision Letter 2]

31 Mar 2026

We are grateful to the Editor and Reviewer #3 for their valuable and thoughtful feedback, which has significantly enhanced the clarity and overall quality of our manuscript. We have thoroughly considered each comment and provide detailed, point-by-point responses below.

Reviewer #3

Comment 1:

Line 24: Expressing the distribution as “0.3% of the country” is not very informative. If the intention is to emphasize the limited distribution of this species, it would be clearer to report the absolute area. Given that this species is mainly restricted to regions of southwest China, providing exact area values would improve clarity.

Response to Comment 1:

Thank you for this helpful suggestion. We agree that expressing the distribution solely as a percentage does not clearly convey the limited extent of the species’ range. Following your recommendation, we have revised the manuscript to include the absolute area of overlap. Specifically, we now report that the highly suitable overlapping area between the two species is approximately 28,800 km², while retaining the percentage (~0.3% of China’s land area) for additional context. This revision provides a clearer representation of the spatial extent and emphasizes the restricted nature of the potential high-risk areas (Lines 23–29 in the Manuscript, the corresponding revisions are highlighted in yellow in the Revised Manuscript with Track).

Comment 2:

Line 43-45: The meaning is somewhat unclear. The term “overlapping ecological niches” is not appropriate here and could be replaced with “co-occurrence”. The statement also conflates climatic effects with niche overlap, as pest–crop interactions already imply spatial co-occurrence; climate primarily modulates their distribution and interaction intensity rather than determining overlap per se.

Response to Comment 2:

Thank you for this insightful comment. We agree that the term “overlapping ecological niches” was not appropriate in this context and that our original wording conflated niche overlap with climatic effects. Following your suggestion, we have revised the text to use “co-occurrence” and clarified the role of climate as a factor that modulates the likelihood of co-occurrence and the intensity of pest–crop interactions, rather than determining overlap itself. (Lines 46–48).

Comment 3:

Line 52: The use of “globally” is inappropriate here; “worldwide” or “across the world” would be more appropriate

Response to Comment 3:

Thank you for this suggestion. We have revised “globally” to “worldwide” in the manuscript (Line 56).

Comment 4:

Line 134-139: Please clarify the source of the data. If a website was used, the full URL and access date should be provided.

Response to Comment 4:

Thank you for this helpful suggestion. We have clarified the data source by providing the full URL along with the access date in the revised manuscript (Line 139): (https://www.tianditu.gov.cn/

, accessed in July 2025).

Comment 5:

Line 204-207: "areas above the threshold were further divided into Low Suitability Areas (LSA), Moderate Suitability Areas (MSA), and High Suitability Areas (HSA). " Please clarify how this classification was performed. Were these categories defined using equal intervals or another thresholding method?

Response to Comment 5:

Thank you for this helpful comment. We have clarified that the classification was performed using an equal-interval approach. Specifically, values above the threshold were divided into three categories (LSA, MSA, and HSA) by equally partitioning the range between the threshold and the maximum suitability value (1). This clarification has been added to the revised manuscript (Lines 208–211).

Comment 6:

Line 214-247: The description of the current potential distribution is overly detailed and somewhat difficult to follow. Listing numerous province names does not necessarily improve clarity, especially for an international readership unfamiliar with China’s administrative regions. I suggest simplifying the presentation by focusing on key patterns rather than exhaustive geographic enumeration. For example, the authors could (1) report the total suitable area, (2) summarize the spatial pattern using broader geographic descriptors (e.g., southwestern, central, or eastern China), and (3) highlight where high-suitability areas are concentrated along with their corresponding area values. This more structured and synthetic description would make the results clearer and more informative.

Response to Comment 6:

We agree that the original description was overly detailed and not sufficiently reader-friendly, particularly for an international audience. Following your recommendation, we have substantially revised this section by removing excessive province-level enumeration and reorganizing the presentation. In the revised version, we now (1) report the total suitable area, (2) summarize the overall spatial patterns using broader geographic descriptors (e.g., southwestern, central, and eastern China), and (3) highlight the distribution and extent of high-suitability areas along with their corresponding area values. These changes improve the clarity, conciseness, and overall readability of the results section.

Comment 7:

Line 296-310: The use of “percentage of China’s land area” is not very informative for assessing ecological or agricultural significance. It would be more meaningful to report absolute area values and/or the proportion of overlap relative to the total suitable area of the host plant. In addition, the comparison based on high-suitability overlap (HSA) is potentially misleading. Pest damage can occur wherever the pest and host co-occur, regardless of suitability class, and areas with moderate suitability may still support populations capable of causing substantial yield loss. I suggest simplifying the analysis by focusing first on the overall spatial overlap between pest and host suitable areas, and then evaluating how this overlap is distributed across the host’s suitability gradient. This would provide a clearer and more ecologically meaningful assessment of potential risk.

Response to Comment 7:

Thank you for this insightful and constructive comment. We agree that using percentages alone is not sufficient to reflect the ecological and agricultural significance of the results. In response, we have revised the manuscript to report absolute area values in place of, and in addition to, percentages.

Following your suggestion, we have also refined the analysis and interpretation by not limiting our focus to high-suitability overlap (HSA). In the revised manuscript, we first describe the general pattern of overlap and then provide a concise interpretation of its potential ecological implications for R. roxburghii. This approach is now reflected in the revised Results (Lines 293–317).

In addition, we have strengthened the Discussion to clarify that overlap in suitable areas represents the potential for co-occurrence rather than direct damage, and therefore should not be interpreted as a direct measure of pest impact. We also explicitly note that most of the overlap occurs within low-suitability areas (LSA), indicating that the host plant still has a large spatial extent where it may be exposed to pest pressure, although the probability of occurrence is relatively low. Furthermore, although the overlap in high-suitability areas (HSA) remains limited, these regions may represent relatively higher-risk zones. (Lines 374–392).

Comment 8:

Line367-371, 411: The interpretation appears to assume that spatial overlap directly reflects pest impact, which is not necessarily valid. As noted above, overlap does not equal impact strength, and this distinction should be more clearly addressed.

Response to Comment 8:

Thank you for this helpful comment. We agree that spatial overlap should not be interpreted as direct pest impact. In the revised manuscript, we have explicitly clarified this point in the Discussion (Lines 374–377), where we state that the term “potential threat” refers to the possibility of spatial co-occurrence between the pest and the host, rather than actual pest damage. This clarification helps to avoid overinterpretation of the overlap results and ensures a more accurate ecological interpretation of the findings.

Comment 9:

The conclusion is overly long and reads more like a condensed repetition of the Results and Discussion sections. It would benefit from a more concise and focused synthesis of the key findings. I suggest restructuring this section to clearly highlight (1) the main objective, (2) two to three key findings, and (3) the primary implication of the study.

Response to Comment 9:

Thank you for this helpful suggestion. We agree that the original conclusion was overly long and largely repetitive of the Results and Discussion sections. Following your recommendation, we have substantially revised and restructured the conclusion to make it more concise and focused.

In the revised version, we first clearly state the main objective of the study, then summarize three key findings in a more structured manner, and finally highlight the primary implication regarding the spatial pattern of potential pest risk under current and future climate scenarios. These revisions improve the clarity and synthesis of the conclusion while avoiding unnecessary repetition of detailed results.

---

## [Decision Letter · Decision Letter 2]

5 Apr 2026

Predicting the Potential Suitable Habitats of Rosa roxburghii and Its Key Pest Grapholita molesta in China Using the MaxEnt Model

PONE-D-25-57528R2

Dear Dr. Hongling Qin,

We’re pleased to inform you that your manuscript has been judged scientifically suitable for publication and will be formally accepted for publication once it meets all outstanding technical requirements.

Kind regards,

Umesh Sharma

Academic Editor

PLOS One

Additional Editor Comments (optional):

Kindly ensure that all minor grammatical errors, equations, and spellings are carefully checked during the proof stage.

Reviewers' comments:

Reviewer's Responses to Questions

**Comments to the Author**

Reviewer #3: (No Response)

2. Is the manuscript technically sound, and do the data support the conclusions?

Reviewer #3: (No Response)

3. Has the statistical analysis been performed appropriately and rigorously?

Reviewer #3: (No Response)

4. Have the authors made all data underlying the findings in their manuscript fully available?

Reviewer #3: (No Response)

5. Is the manuscript presented in an intelligible fashion and written in standard English?

Reviewer #3: (No Response)

Reviewer #3: (No Response)

.

Reviewer #3: No

---

## [Editor Report · Acceptance letter]

PONE-D-25-57528R2

PLOS One

Dear Dr. Qin,

I'm pleased to inform you that your manuscript has been deemed suitable for publication in PLOS One. Congratulations! Your manuscript is now being handed over to our production team.

Kind regards,

on behalf of

Dr. Umesh Sharma

Academic Editor

PLOS One